# Syncope in the Emergency Department: A Practical Approach

**DOI:** 10.3390/jcm13113231

**Published:** 2024-05-30

**Authors:** Ludovico Furlan, Giulia Jacobitti Esposito, Francesca Gianni, Monica Solbiati, Costantino Mancusi, Giorgio Costantino

**Affiliations:** 1Department of Clinical Sciences and Community Health, University of Milan, 20122 Milan, Italy; ludovico.furlan@unimi.it (L.F.); monica.solbiati@unimi.it (M.S.); giorgio.costantino@unimi.it (G.C.); 2Internal Medicine Department, IRCCS Cà Granda Ospedale Maggiore Policlinico, 20122 Milan, Italy; 3Emergency Medicine School, Department of Advanced Biomedical Science, University of Naples Federico II, 80138 Naples, Italy; juliajaco19@hotmail.com (G.J.E.); costantino.mancusi@unina.it (C.M.); 4Emergency Department, IRCCS Cà Granda Ospedale Maggiore Policlinico, 20122 Milan, Italy

**Keywords:** syncope, management reasoning, orthostatic hypotension, arrhythmia, risk stratification

## Abstract

Syncope is a common condition encountered in the emergency department (ED), accounting for about 0.6–3% of all ED visits. Despite its high frequency, a widely accepted management strategy for patients with syncope in the ED is still missing. Since syncope can be the presenting condition of many diseases, both severe and benign, most research efforts have focused on strategies to obtain a definitive etiologic diagnosis. Nevertheless, in everyday clinical practice, a definitive diagnosis is rarely reached after the first evaluation. It is thus troublesome to aid clinicians’ reasoning by simply focusing on differential diagnoses. With the current review, we would like to propose a management strategy that guides clinicians both in the identification of conditions that warrant immediate treatment and in the management of patients for whom a diagnosis is not immediately reached, differentiating those that can be safely discharged from those that should be admitted to the hospital or monitored before a final decision. We propose the mnemonic acronym RED-SOS: Recognize syncope; Exclude life-threatening conditions; Diagnose; Stratify the risk of adverse events; Observe; decide on the Setting of care. Based on this acronym, in the different sections of the review, we discuss all the elements that clinicians should consider when assessing patients with syncope.

## 1. Introduction

Syncope is defined by the European Society of Cardiology (ESC) as a transient loss of consciousness (TLOC) due to cerebral hypoperfusion, characterized by a rapid onset, short duration, and spontaneous complete recovery [1]. 

Syncope is a condition frequently encountered in clinical practice, affecting up to 30% of the general population in their lifetime and being the cause of 0.6–3% of all emergency department (ED) visits [1]. ED physicians deal with several challenges in the management of patients with syncope. As syncope may be the expression of many diseases, either benign or potentially fatal, it is crucial to identify which patients may benefit from immediate treatment and further testing. However, since most patients are asymptomatic at the time of first evaluation, diagnosis of the underlying cause is often difficult. It is estimated that only 50–60% of episodes end up with a definitive diagnosis at the time of the first evaluation in the ED [2]. The absence of a uniform approach generates a high variability in the management of individual patients and, even with the suspicion of a specific etiology, diagnostic tests, treatments, and procedures significantly differ from one patient to another. Moreover, the rate of hospital admissions following ED evaluation varies greatly across countries, ranging from less than 20% in Canada to more than 80% in the United States [3,4,5].

In this context, rather than striving to reach a specific diagnosis, we believe that efforts should be directed toward finding a standardized management strategy. This may help with tailoring care to the specific patient’s needs. 

In this review, we propose as a practical tool the acronym **RED-SOS: R**ecognize syncope; **E**xclude life-threatening conditions; **D**iagnose; **S**tratify the risk of adverse events; **O**bserve; decide on the **S**etting of care (Figure 1).

The proposed management pathway is intended to be used in adult patients. Discussion of the management of syncope in children is outside the scope of the current review and has been analyzed by publications focusing on specific conditions such as inherited arrhythmia in children [6]. 

## 2. Recognize Syncope

When approaching a patient with TLOC, the first step is to recognize which patients have had a real syncopal event. Seizures and falls are the main conditions that can mimic syncope and must be excluded. A description of the episode by the patient or by bystanders is essential in this phase.

### 2.1. Syncope vs. Falls 

The differential diagnosis between falls and syncope is usually straightforward in patients who can clearly recall the dynamic of the episode. Symptoms and signs preceding or following the fall, such as lightheadedness, palpitations, chest discomfort, bilateral lower limb weakness and pallor, usually suggest syncope; conversely, palmar bruises may indicate falls. Importantly, the correct identification of falls is usually straightforward in young patients but may be more challenging in the elderly. Indeed, falls, as consequences of mechanical slips or trips, are particularly frequent in this age group, but patients may not clearly recall the dynamic. At the same time, the prevalence of syncope increases with age, and syncope in older patients is more frequently associated with cardiac etiologies, major trauma, and short-term adverse events [7]. Therefore, as a practical rule, falls in the elderly should be managed as syncopal episodes until clearly proven otherwise. Of note, it is essential to consider that the threshold at which we accept the possibility of missing a syncopal episode varies from case to case. It may be lower (i.e., accepted risk of missing a syncope higher) for those who are not candidates for invasive treatments, such as older patients with dementia and significant motor impairment. On the contrary, the threshold may be higher (i.e., accepted risk of missing syncope lower) for those in which a syncopal episode would have major consequences due to working conditions, either for the patient or other people (i.e., construction workers, public transport drivers, etc.). 

### 2.2. Syncope vs. Seizures

Seizures may also resemble syncope and cause TLOC. Between 20 and 30% of patients with a diagnosis of seizures may have had syncope [8,9]. 

Table 1 reports the characteristics with proven good accuracy in discriminating between syncope and seizures [10,11,12]. The accurate description of the episode, together with the signs and symptoms, are usually sufficient to discriminate these two entities. In addition, a past medical history of previous epileptic episodes or the presence of predisposing factors such as cerebral lesions increases the likelihood of seizures. Lateral tongue biting, slow neurological recovery following TLOC (post-critical state) and an increase in serum lactate and creatine kinase appear to be the single factors with the highest accuracy in discriminating epilepsy and syncope. Indeed, lateral tongue biting appears to be the single most accurate sign for confirming the diagnosis of epilepsy, with a reported specificity of 96% and good positive likelihood ratio (8.17, 95% CI 2.97–22.46); nevertheless, this comes at the cost of a poor sensitivity (33%) [13]. 

Serum creatine kinase and lactate may also help clinicians in differentiating between syncope and epilepsy. An increase in the creatine kinase levels has shown a good overall specificity (between 85 and 100%) but a reported sensitivity between 12 and 75%. The serum lactate specificity and sensitivity can vary, respectively, between 87 and 97% and 73 and 88%, depending on the clinical setting in which they are performed [14]. Several scores have been proposed to distinguish syncope from epilepsy based on historical criteria [15,16]; however, only a few validation studies have been conducted throughout the years to support their use in clinical practice [17].

Of note, neither EEG nor neurologic imaging (MRI, CT scan or vascular ultrasound) have any additional clinical benefit for a differential diagnosis over an accurate history collection [18,19].

### 2.3. Syncope vs. Other Rare Forms of TLOC

Other less common conditions to include in the differential diagnosis of syncope are non-syncopal rare causes of TLOC such as subclavian steel syndrome, traumatic TLOC, or psychogenic TLOC (e.g., psychogenic “pseudo-syncope” or “pseudo-seizures”). TLOC during conversion disorders in the young may be misinterpreted as syncope and should be suspected in cases of prolonged LOC (>5 min), closed eyes or resisted eyelid opening during the episodes or a fluctuating state of consciousness. In such cases, paradoxically, clinicians should be more interested in not missing a diagnosis of conversion syndrome rather than syncope. Indeed, in the latter case, the presumptive diagnosis would often be that of benign neuro-mediated syncope, and no further tests or treatments would be needed. On the contrary, conversion syndromes may benefit from dedicated care and support. 

Pre-syncope (i.e., a sensation of impending syncope without LOC) is another condition that can be misinterpreted as syncope. Since the definition of pre-syncope is even more vague and less standardized than that of syncope [20], and the prognostic implications are less studied [21,22], we believe that it should not be managed as syncope. 

## 3. Exclude Life-Threatening Conditions

Once it has been established that syncope is more likely than other forms of TLOC, the clinician should determine whether there is a potentially life-threatening condition and provide immediate care or second-level diagnostic assessments. Of note, even before the Recognize phase, patients with clear signs and symptoms of clinical instability or shock should be immediately managed according to the institution’s dedicated flowchart (such as the American Heart Association Advanced Cardiovascular Life Support algorithms [23]). The real challenge for clinicians resides in the management of patients that are completely or relatively stable. In this context, the goals should be to identify which patients are at risk of severe conditions that could be diagnosed with quickly available bedside exams and would benefit from further diagnostic tests and urgent treatments. The main conditions to consider in this phase are the following: acute aortic syndromes (AASs), acute coronary syndromes (ACS), pulmonary embolism (PE), life- threatening arrhythmia, uncontrolled hemorrhages, and subarachnoid hemorrhage (SAH). We suggest, as a guide for the ED physician, focusing on the ECG findings and on the identification of red flags.

### 3.1. ECG Findings

ECG is mandatory for every patient to identify the signs of acute myocardial infarction and major arrhythmias that may need prompt interventions, i.e., third-degree AV block, wide complexes tachycardia and extremely fast narrow complex tachycardia (heart frequency > 160 bpm) or PM/ICD malfunction.

### 3.2. Red Flags and Life-Threatening Conditions

The red flags that should be considered at this stage include dyspnea, pain (headache or thoraco-abdominal), disability, persistent tachycardia, and hypotension. The presence of any of these red flags should guide further tests, such as d-dimer, troponin testing, heart ultrasound or computed tomography. Management of specific diseases is beyond the objective of this review; therefore, we will discuss only a few relevant topics. 

Only 13% of patients with acute aortic dissection (AAD) present with syncope [24], while more than 80–90% present with acute chest pain [25,26]. Considering the overall low prevalence of AAD among ED visits (1 case every 12,000 visits in the US [26]), syncope in the absence of other symptoms is extremely rare in AASs. The integration of a low risk according to the Aortic Dissection Detection Risk Score (i.e., score ≤ 1) and negative D-dimer (i.e., <500 ng/mL) has shown promising results in excluding AASs [27,28]. 

PE is a potentially life-threatening cause of syncope; however, we believe that a small proportion of patients would benefit of further diagnostic tests to exclude PE. Data on patients hospitalized for syncope have suggested that up to 17% (95% CI, 14–21) have evidence of PE upon CT scan when managed through an algorithm to exclude PE, independently of any other potential identified cause of syncope [29]. Yet, there is no evidence that the extensive application of diagnostic pathways to exclude PE in all syncopal episodes would provide any clinical benefit to patients in terms of a better prognosis. Furthermore, data on the medium- and long-term outcomes suggest that the overall prevalence may be less than 1% of patients with syncope [30,31]. Thus, we believe that PE should be considered in all patients presenting with syncope, but also that not all patients with syncope should undergo tests to exclude PE. 

An abrupt and extremely severe headache may be the presenting symptom of SAH. Even if variably included by guidelines among the syncopal forms of TLOC [1], we believe that SAH should be considered among the severe causes of syncope [1]. Indeed, the LOC in SAH is likely caused by a global cerebral hypoperfusion induced by a sharp increase in the intracranial pressure [32]. 

Focal neurologic signs causing disability may also be present in aortic and main branch dissection and intracranial hemorrhages. 

## 4. Diagnose and Stratify the Risk of Adverse Events

After excluding life-threatening conditions, the following step should involve making a presumptive diagnosis and stratifying the risk of adverse events in the short term. Syncope is a unique condition in which a definitive diagnosis can only be made at the precise time of the loss of consciousness. Even if some patients may experience multiple syncopal episodes, it is highly unlikely that this happens during the clinical evaluation. 

From a management perspective, it is thus essential to contemporarily provide a potential diagnosis and to stratify the risk of adverse events (Table 2 and Table 3). After the initial evaluation of patients with syncope, the decision to perform second-level diagnostic tests, treatments or procedures should be tailored to the specific patient. 

Once a likely diagnosis and risk stratification are performed, patients usually fall in to one of the following three groups [33]: -Likely benign cause of syncope and low-risk features, with no high-risk features. -Likely severe cause of syncope and high-risk features. Patients should be admitted for further testing. -All other forms. This group includes:○patients with a presumptive benign cause of syncope but with high-risk comorbidities;○patients with a potential worrisome mechanism of syncope but no high-risk comorbidities;○patients with neither low-risk nor high-risk features.

The presumptive diagnosis and risk stratification rely on pieces of information that the clinician obtains from the medical history, the description of the episode provided by the patient or bystanders, the clinical examination, the ECG findings and the measurement of the supine and standing blood pressure to evaluate the presence of orthostatic hypotension (Table 2 and Table 3). All such factors should be considered collectively, but they are presented in different sections here for better clarity.

### 4.1. History and Episode Characteristics

Several elements of the patient’s history and the characteristics of the syncopal event have been shown throughout the literature to be related to either benign or malignant forms of syncope. In a recent meta-analysis, the most accurate factors to exclude a cardiac syncope were age < 35 years old (LR, 0.13 [95% CI, 0.06–0.25]) and mood change or prodromal preoccupation before syncope (LR, 0.09 [95% CI, 0.02–0.38]) [34]. In another analysis of seven studies on syncope conducted in four different countries, age < 40 years old had the best negative likelihood of excluding a cardiac syncope [35]. Indeed, in the cohort of interest, less than 1% of patients with cardiac syncope were less than 40 years old [35]. Conversely, a history of structural heart disease had the highest positive likelihood ratio for cardiac syncope, with a prevalence of 87% in this group of patients. In the same study, nausea, diaphoresis, long prodromes and blurred vision were more common in non-cardiac syncope, even if less predictive than a younger age. An age greater than 60 years old, male gender, syncope occurring during physical activity, with multiple closely related episodes, or in the supine position were more frequently related to cardiac syncope [35].

### 4.2. Physical Examination

The cardiac exam should focus on signs that point to the possibility of structural heart disease, such as rubs, gallops and murmurs, and therefore may warrant ultrasound testing [1]. Evidence of trauma upon physical examination may prompt further radiologic investigations to exclude fractures or bleedings. A head CT scan should be considered only with the suspicion of SAH (severe headache) or in the case of significant head trauma due to a fall. All patients should undergo supine and standing blood pressure measurement to verify the presence of orthostatic hypotension (OH), i.e., a drop in the systolic/diastolic blood pressure of at least 20/10 mmHg or systolic blood pressure below 90 mmHg while standing. OH may be caused by acute conditions affecting the circulating blood volume (bleeding, severe dehydration from vomit, fever, or diarrhea), the vasomotor activity (sepsis) or both. Management thus depends on the underlying condition. In the case of bleeding, global hypotension only occurs when more than 30% of the blood volume is lost. Nevertheless, OH may occur with less significant losses. OH may also be caused by chronic conditions such as neurodegenerative diseases, diabetes, hypertension, cancer and medications such as diuretics, beta-blockers, vasodilators, phenothiazines, antidepressants, SGLT-2 inhibitors and many others [36]. Patients experiencing syncope due to these latter forms of OH usually have a benign short-term prognosis. However, it is important to note that, while OH can be reasonably considered responsible for the syncopal episode when an acute etiology is identified, the presence of drug-induced or chronic disease-associated OH does not exclude other concomitant causes of syncope. Indeed, the prevalence of OH may be as high as 30% in patients older than 70 years old [37], who are at the same time at an increased risk of cardiac syncope. Thus, in the absence of documented syncope while standing, or a highly suggestive history, the isolated identification of OH does not provide a definitive diagnosis. Furthermore, symptoms suggestive of cardiac syncope or high-risk features should warrant management independently of the presence of OH with no symptoms while standing.

The ESC guidelines recommend that any patients older than 40 years old with syncope of unknown origin compatible with a reflex mechanism receive carotid sinus massage (CSM) [1]. Carotid sinus hypersensitivity is defined as a ventricular pause lasting >3 s and/or a fall in systolic BP of >50 mmHg observed during carotid sinus massage (CSM). Carotid sinus syndrome (CSS) refers to the induction of syncope during CSM and carotid sinus hypersensitivity. CSS can be further classified in the cardio-inhibitory, vaso-depressive or mixed type depending on the presence of one criterion or both criteria for carotid sinus hypersensitivity. The rate of complications during CSM, mainly TIA or stroke, is extremely low (<0.5%) and most episodes do not result in permanent neurologic sequelae [38,39,40,41], although caution should be maintained in patients with carotid bruits. Nevertheless, the net clinical benefit of CSM in patients with syncope is still debated [1]. Indeed, while it has been demonstrated that cardio-inhibitory forms of CSS may predict the risk of future asystolic pauses [42], there is no adequate evidence that pacemaker implantation has any benefit [43,44], while there is low-quality evidence that it may have no benefit at all [45]. Thus, we believe the decision to perform a CSM in a patient with syncope should depend on whether the clinician would consider pacemaker implantation for the patient. 

### 4.3. ECG, Heart Ultrasound, and Other Tests

Other than immediately life-threatening conditions, some ECG findings are diagnostic for cardiac arrhythmic syncope, while others are suggestive of arrhythmic or structural heart syncope and thus should be included in the risk stratification of the patient (Table 2 and Table 3). In patients with a cardiac pacemaker (PM) or intra-cardiac defibrillator (ICD), device interrogation should be performed to identify possible arrhythmic causes of the syncopal episode. Patients should then be managed accordingly to the specific arrhythmia.

All patients with suspected cardiac syncope should undergo heart ultrasound to look for structural heart diseases. Quick point-of-care ultrasound (POCUS) may be a good screening option to look for an abnormal aortic valve gradient or increase > 2 m/s in the peak velocity, pericardial fluid, right ventricle dilation and reduction of left ventricular global function, or to identify other rare conditions, such as cardiac myxomas [46,47].

Routine blood tests are rarely informative as to the nature of the syncopal episode but might be appropriate in non-low-risk patients to evaluate the overall frailty in selected cases. 

### 4.4. Risk-Stratification Tools

The factors at play for risk stratification have been already described and are reported extensively in Table 3. We support the use of such factors to identify patients at high, low or neither high or low risk of adverse events.

Several tools to stratify patients with syncope in the ED have been proposed throughout the literature [48,49,50,51,52,53,54,55,56,57]. However, from a patient management perspective, the risk stratification scores do not seem to perform better than clinical judgement in predicting short-term adverse events in patients with syncope [58,59,60]. 

### 4.5. Low-Risk Patients

Patients with low-risk features and presumptive benign causes of syncope can be safely discharged due to the overall very low risk of adverse events. Reassurance as to the benign nature of the episode should be provided and patients should be informed about how to recognize prodromal symptoms and to avoid trauma related to the loss of consciousness. Moreover, in patients < 65 years old, physical counterpressure maneuvers by means of limbs isometric contraction or squatting have been shown to increase the blood pressure and may abort impending vasovagal syncope [61,62,63,64]. The efficacy may be less in syncope of unknown cause [61,62]. Salt (6–10 g of sodium chloride/day) and water supplementation (at least 500 mL above the median water input) have also been shown to provide some benefit as prophylaxis measures to avoid vasovagal or orthostatic forms of syncope [65,66,67]. In some circumstances, rapid fluid infusion provides symptom relief in low-risk patients. 

Pharmacologic treatment of specific forms of syncope is outside the scope of the present review, and it usually requires wide expertise in the field. 

While it is intuitive how the use of anti-hypertensive medications may contribute to syncope due to orthostatic hypotension, particularly in the elder population [68,69], a recent individual patient data metanalysis suggested that intensive blood pressure treatment guarantees overall mortality benefit, even among those with standing hypotension [70]. A personalized decision should be made in the setting of hypertensive patients with syncope due to orthostatic hypotension, carefully balancing the mitigation of cardiovascular risk and the risk of falls and trauma. 

### 4.6. High-Risk Patients

Due to the risk of adverse event and the high likelihood of needing further testing, such patients should be admitted to the hospital in units with 24 h ECG telemetry available and capability to manage clinical deterioration. 

## 5. Observe

Patients considered neither low nor high risk may benefit from a period of observation that may be helpful for a subsequent re-classification into high or low risk. Observation should be performed in the ED or in a dedicated setting with resuscitation facilities available, and it seems to be an adequate alternative to routine hospitalization for patients with unexplained syncope [71]. During the observation phase, patients should be monitored with ECG recording for arrhythmia and should undergo heart ultrasound, if not performed before. SCM may be helpful in a very limited proportion of patients with recurrent syncope of suspected reflex etiology (see the dedicated paragraph for details). 

The rationale by which monitoring may provide a benefit in patients with syncope is the proximity in time between the syncopal episode and adverse events. Indeed, most short-term adverse events in patients with syncope tend to present in the first three days after the ED evaluation and are mainly brady- or tachy-arrhythmias [72,73]. However, it is still unclear what is the optimal duration of ECG monitoring. In a prospective study that included 5581 patients evaluated in the ED for syncope and risk-stratified with the Canadian Syncope Risk Score, up to 50% of all arrhythmic episodes occurred within 2 h in low-risk patients and within 6 h in medium- and high-risk patients after ED arrival [73]. In another prospective study on 242 non-low-risk patients with syncope, cardiac monitoring for more than 12 h showed a good sensitivity for 7-day adverse events (0.89, 95% CI: 0.65 to 0.99) [74]. 

Though definitive recommendations on the optimal duration of cardiac monitoring cannot be provided, we suggest 12 h in most patients. A shorter duration may be considered for low-risk patients with an unknown mechanism of syncope. 

## 6. Setting of Care

The net benefit of hospitalization in syncope strictly depends on the patient’s risk of adverse events and on the goals of hospitalization. 

Low-risk patients do not usually require additional diagnostic tests and can safely be discharged from the ED. If any additional evaluation is thought necessary for comfort, therapy, or counseling, the patient should be treated as an outpatient in a syncope clinic, syncope unit, or specialty clinic. Hospitalization in the low-risk group of patients provides very limited additional diagnostic benefit and may be associated with an up to 13% absolute risk of in-hospital adverse events [75]. 

Accordingly, a few studies have shown how the overall rate of hospitalization for syncope has decreased in recent years, with no apparent risk of revisits, highlighting once again the overall safety of discharge for the majority of patients [76,77].

High-risk patients are probably those who benefit the most from hospitalization, both in terms of the diagnostic accuracy and the prevention of adverse events. Nevertheless, to the best of our knowledge, no specific study in this group of patients has been performed so far.

It is not clear whether hospitalization provides any additional benefit in terms of the prognosis in patients with syncope. The STePS study [78] has, indeed, provided evidence that hospitalization may positively affect the short-term but not the long-term (1 year) prognosis. On the other hand, the risks related to hospitalization have been extensively described throughout the literature, and a recent systematic review has estimated a 6% overall risk of preventable harm at the time of patient admission [79].

Goals other than defining the etiology or preventing adverse events may influence the decision to admit or discharge a patient. Consequences related to the trauma derived from syncope, such as fractures and head trauma, may require in-hospital treatment or observation. In other cases, patients are admitted because of the underlying disease that has caused a benign cause of syncope, as happens in orthostatic hypotensive syncope during sepsis. Social factors (loneliness and inadequate social support) and overall frailty, particularly in older patients, should also be considered, particularly when the cause of syncope is itself benign but the patient may experience adverse events due to a fall. 

Thus, as a general rule, when deciding to admit a patient, it should be clear to the ED physician what the goal of hospitalization is for the specific patient and choices should be made accordingly. On the other hand, for those discharged from the ED, it is essential to follow-up the patient in dedicated units or ambulatory care facilities. 

## 7. Conclusions

Syncope is an elusive syndrome in which the exact diagnosis of the underlying cause is frequently impossible. Management strategies rather than simplistic diagnostic pathways are needed as a response to this diagnostic uncertainty. Here, we propose the RED-SOS acronym (**R**ecognize; **E**xclude; **D**iagnose; **S**tratify the risk of adverse events; **O**bserve; **S**etting of care) as a mnemonic tool to recall all the essential steps when dealing with patients with syncope in the ED. To exclude potential life-threatening conditions, physicians should focus on red flag signs or symptoms (dyspnea, hypotension, tachycardia, disability, severe headache, thoraco-abdominal pain). We believe that the proposed stepwise approach could provide physicians with adequate information to define “how” and “in which setting” patients should be treated or monitored. Rather than focusing on simple risk stratification through the available tools, efforts should be directed toward the management of the patient risk. 

## Figures and Tables

**Figure 1 jcm-13-03231-f001:**
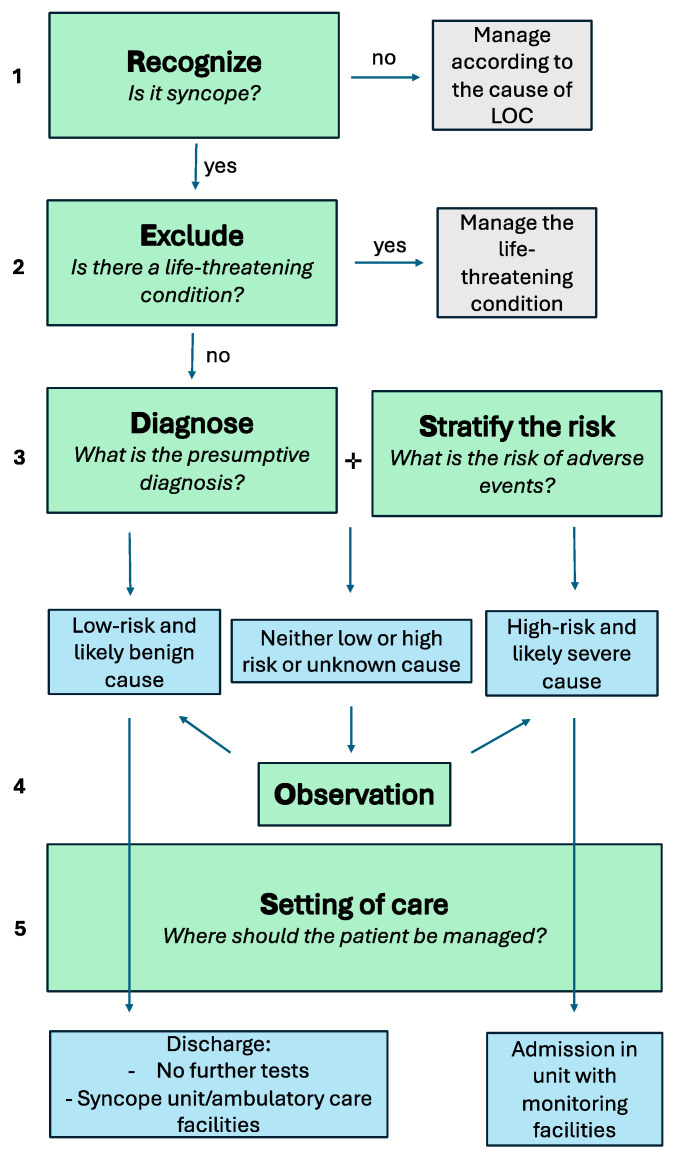
RED-SOS management flowchart.

**Table 1 jcm-13-03231-t001:** Differential diagnosis between syncope and epilepsy.

Clinical Characteristics	Syncope	Epileptic Seizures
Aura	Not present.	May be experienced in terms of an unpleasant smell, tastes, or a rising feeling of sickness. In recurrent episodes, patients may recognize symptoms that precede spells.
Lightheadedness	Common.	Uncommon.
Skin color during TLOC	Pallid.	Cyanotic.
Tongue biting	Rare. Usually at the tip of the tongue.	Common. Lateral side of the tongue.
Eye deviation	May be present. Usually an upward conjugated gaze.	May be present. Usually a lateral conjugated gaze.
Duration of movements during TLOC	A few seconds, if present.	Up to minutes.
Jerks and movements	If present, follows TLOC.	May precede or be simultaneous with TLOC.
Duration of TLOC	Few seconds.	Up to minutes.
Recovery	Rapid with no neurologic sequelae.	Slower, complete recovery may take hours. TLOC may be followed by stupor, sopor and/or transient neurologic motor deficits (Todd’s paralysis).
Creatine Kinase and/or Lactate increase	Absent.	Frequent.

**Table 2 jcm-13-03231-t002:** The diagnostic electrocardiographic (ECG) features of patients with cardiac arrhythmic syncope.

Diagnostic ECG Findings
Pause (>3 s)
Sustained or non-sustained ventricular tachycardia, whether symptomatic or asymptomatic
Rapid paroxysmal supraventricular tachycardia
High-grade AV block (Mobitz II-second- and third-degree AV block)
Bradycardia (<30 b.p.m.), whether symptomatic or asymptomatic
Bradycardia (<50 b.p.m.) in a symptomatic patient
Tachycardia (>120 b.p.m.) in a symptomatic patient
Dysfunction of an implantable cardiac device (pacemaker or ICD)
Alternating left and right BBB

**Table 3 jcm-13-03231-t003:** Features identifying low- and high-risk patients.

	**Low Risk**	**High Risk**
Syncope features	■ Prodromes (vomiting, feeling warm, and sweating) ■ Triggers (pain, emotions, cough, defecation, micturition) ■ Following extended standing or in hot, crowded areas ■ Standing from supine/sitting position ■ Either during or after a meal	■ Sudden onset of chest discomfort, dyspnea, headache, or abdominal pain ■ Palpitations as only prodrome ■ In a supine position or during exertion
Personal and family history	■ Young age (<40 years) ■ Extended duration (years) of syncope exhibiting the same traits as the current episode ■ No history of structural heart disease	■ Family history of sudden death (>at a young age) ■ Severe heart artery disease, either coronary or structural (e.g., heart failure, previous myocardial infarction, or ICD implantation, reduced left ventricular ejection fraction)
First evaluation findings	■ No abnormalities in physical evaluation	■ Systolic blood pressure finding in the ED of ≤90 mmHg ■ Bradycardia (heart rate ≤ 40 bpm) ■ Anemia (hemoglobin <9 g/dL)
ECG findings		■ New (or previously unknown) left BBB ■ Bifascicular block ■ Brugada ECG pattern ■ ECG changes consistent with acute ischemia ■ Non sinus rhythm (new) ■ Pre-excited QRS complexes ■ Negative T waves in right precordial leads, epsilon waves suggestive of arrhythmogenic cardiomyopathy ■ Prolonged QTc (>450 ms)

## Data Availability

No new data were created or analyzed in this study. Data sharing is not applicable to this article.

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
