# Peer review of "Syncope in the Emergency Department: A Practical Approach"

_jcm, 2024, doi:10.3390/jcm13113231_

Round 1
Reviewer 1 Report
Comments and Suggestions for Authors
That´s a very interesting approach to the subject. I believe that this paper may be very useful in the clinical practice.
My few suggestion for you are:
1- You need to decide if you will use, for "transient loss of consciousness, the acronym T-LOC or TLOC (as defined in the Introduction section), as both of them were used.
2- The numerations of the sections are not well organised. Introduction is the number 1, but then the number 2 is the first step of your proposed algorithm (recognise syncope) and the Conclusions section has no number. It needs a better organisation.
Author Response
Dear Editor,
we would like to thank the reviewer for her/his comments.
Please find below our response to the reviewer comments.
Reviewer comments:
“That´s a very interesting approach to the subject. I believe that this paper may be very useful in the clinical practice.
My few suggestion for you are:
1- You need to decide if you will use, for "transient loss of consciousness, the acronym T-LOC or TLOC (as defined in the Introduction section), as both of them were used.
We would like to thank the reviewer for his/her comment. We have modified the manuscript using only TLOC acronym.
2- The numerations of the sections are not well organised. Introduction is the number 1, but then the number 2 is the first step of your proposed algorithm (recognise syncope) and the Conclusions section has no number. It needs a better organisation.
We apologize for the lack of consistency and thank the reviewer for his/her comment. We modified the organization of the manuscript using numbers only for the different sections of the proposed acronym RED-SOS. We have also added, as suggested by reviewer 2, a figure with a flowchart of the proposed acronym.

Reviewer 2 Report
Comments and Suggestions for Authors
Congratulations to authors for the interesting topic. I have some comments that could help to improve the paper and make it suitable for publication:
The article's structure could be more streamlined to avoid repetition, especially in sections discussing the differentiation between syncope and seizures.
Authors should improve the transitions between different sections could be smoother. Moreover some sections, like Exclude life-threatening conditions and "Recognize syncope" could benefit from clearer subheadings and a more logical progression.
I strongly suggest to include a pediatric section with a focus on epidemiology of different syndromes that can cause syncope in this category of patient (authors can use Inherited Arrhythmias in the Pediatric Population: An Updated Overview. Medicina (Kaunas). 2024 Jan 3;60(1):94. doi: 10.3390/medicina60010094. PMID: 38256355; PMCID: PMC10819657.)
The section regarding the distinction between syncope and epilepsy could be resumed in order to highlight the most relevant diagnostic criteria.
Consider adding visual aids like flowcharts or diagrams could help illustrate complex diagnostic pathways and management strategies.
All in all I believe that this paper will be suitable for publication after following all of this recomendations.
Comments on the Quality of English LanguageTry to reduce the complexity of phrases such as: "In this context, rather than striving to reach a specific diagnosis we believe there is a need of a standardized management strategy that helps tailoring care on the specific patient’s needs."
Author Response
Dear Editor,
Please find below our response to the reviewer comments.
Reviewer comments: “Congratulations to authors for the interesting topic. I have some comments that could help to improve the paper and make it suitable for publication:
- The article's structure could be more streamlined to avoid repetition, especially in sections discussing the differentiation between syncope and seizures.
We would like to thank the reviewer for her/his/their comments. We have reduced the information provided in the manuscript, that have been extensively reported in table 1.
- Authors should improve the transitions between different sections could be smoother.
We would like to thank the reviewer for her/his/their comments. We have re-organized the paragraphs numbers, as suggested by reviewer n.1, to improve coherence among the sections. However, we also feel that the different stages of proposed acronym should clearly distinguished to reduce possible confusion. We also have included as suggested a flowchart in figure 1 to improve clearness.
- Moreover some sections, like Exclude life-threatening conditions and "Recognize syncope" could benefit from clearer subheadings and a more logical progression.
We thank the reviewer for her/his/their comment. We modified the structure of the manuscript including subheadings as suggested.
- I strongly suggest to include a pediatric section with a focus on epidemiology of different syndromes that can cause syncope in this category of patient (authors can use Inherited Arrhythmias in the Pediatric Population: An Updated Overview. Medicina (Kaunas). 2024 Jan 3;60(1):94. doi: 10.3390/medicina60010094. PMID: 38256355; PMCID: PMC10819657.)
We thank the reviewer for her/his/their comment. We agree that inclusion of data on children management would be of great interest. However, we believe such an issue would need a throughout and extensive discussion and probably a dedicated review.
We have added a sentence in which we recognize the importance to distinguish the management of syncope in adult and children at the beginning of the manuscript, including the proposed citation.
- The section regarding the distinction between syncope and epilepsy could be resumed in order to highlight the most relevant diagnostic criteria.
We would like to thank the reviewer for his/her/their comment. As suggested, we have reduced the amount of information reported, that can be found in table 1.
- Consider adding visual aids like flowcharts or diagrams could help illustrate complex diagnostic pathways and management strategies.
We would like to thank the reviewer for his/her/their comment. We have included a figure that summarizes the proposed management flowchart.
All in all I believe that this paper will be suitable for publication after following all of this recomendations.
Comments on the Quality of English Language
Try to reduce the complexity of phrases such as: "In this context, rather than striving to reach a specific diagnosis we believe there is a need of a standardized management strategy that helps tailoring care on the specific patient’s needs."
We thank the reviewer for his/her/their comment and have revised the manuscript. We modified the sentence.

Reviewer 3 Report
Comments and Suggestions for Authors
Furlan L et al. have proposed the acronym RED-SOS (Recognize; Exclude; Diagnose; Stratify risk of adverse events; Observe; Setting of care) for syncope. 
The manuscript is well written and well algorithmic in how it approaches syncope.
The clinical benefits of using this RED-SOS algorithm will hopefully be demonstrated in future studies.
Minor
# I could not find Table 4 in this manuscript.
Author Response
Dear Editor,
Please find below our response to the reviewer comments.
Reveiwer comments:
Furlan L et al. have proposed the acronym RED-SOS (Recognize; Exclude; Diagnose; Stratify risk of adverse events; Observe; Setting of care) for syncope. 
The manuscript is well written and well algorithmic in how it approaches syncope.
The clinical benefits of using this RED-SOS algorithm will hopefully be demonstrated in future studies.
Minor
# I could not find Table 4 in this manuscript.
We apologize for the error and would like to thank the reviewer for her/his/their comment. There is no table 4, we were referring to table 3 instead and, thus, modified the manuscript accordingly.
